# The human auditory brainstem response to running speech reveals a subcortical mechanism for selective attention

**Antonio Elia Forte, Octave Etard, Tobias Reichenbach***

Department of Bioengineering, Centre for Neurotechnology, Imperial College London, London, United Kingdom

**Abstract** Humans excel at selectively listening to a target speaker in background noise such as competing voices. While the encoding of speech in the auditory cortex is modulated by selective attention, it remains debated whether such modulation occurs already in subcortical auditory structures. Investigating the contribution of the human brainstem to attention has, in particular, been hindered by the tiny amplitude of the brainstem response. Its measurement normally requires a large number of repetitions of the same short sound stimuli, which may lead to a loss of attention and to neural adaptation. Here we develop a mathematical method to measure the auditory brainstem response to running speech, an acoustic stimulus that does not repeat and that has a high ecological validity. We employ this method to assess the brainstem's activity when a subject listens to one of two competing speakers, and show that the brainstem response is consistently modulated by attention.

DOI: https://doi.org/10.7554/eLife.27203.001

## Introduction

It is well known that selective attention to one of several competing acoustic signals affects the encoding of sound in the auditory cortex (*Shinn-Cunningham, 2008*; *Hackley et al., 1990*; *Choi et al., 2013*; *Fritz et al., 2007b*; *Hillyard et al., 1973*; *Womelsdorf and Fries, 2007*; *Fritz et al., 2007a*; *Näätänen et al., 2001*). Because extensive auditory centrifugal pathways carry information from central to more peripheral levels of the auditory system (*Winer, 2006*; *Pickels, 1988*; *Song et al., 2008*; *Bajo et al., 2010*), neural activity in the subcortical structures may contribute to attention as well. Previous attempts to determine an attentional modulation from recording the auditory brainstem response through scalp electrodes have, however, yielded highly inconclusive results.

In particular, one investigation found that selective attention alters the brainstem's response to the fundamental frequency of a speech signal (*Galbraith et al., 1998*), while another study concluded that this response is modulated in an unsystematic but subject-specific manner (*Lehmann and Schönwiesner, 2014*) and a third recent experiment did not find a significant attentional effect (*Varghese et al., 2015*). Results on the effects of attention on the auditory-brainstem response to short clicks or pure tones are similarly inconclusive (*Brix, 1984*; *Gregory et al., 1989*; *Hoormann et al., 2000*; *Galbraith et al., 2003*). These inconsistencies may result from a main experimental limitation in these studies: because the brainstem response is tiny, its measurement requires hundred- to thousandfold repetition of the same sound. The large number of repetitions may lead to difficulties for subjects in sustaining selective attention, to adaptation in the nervous system, and to a reduction in efferent feedback (*Lasky, 1997*; *Kumar Neupane et al., 2014*).

To overcome this limitation, we develop here a method to measure the auditory brainstem's response to natural running speech that does not repeat. We then use this method to assess the

*For correspondence:
reichenbach@imperial.ac.uk

**Competing interests:** The authors declare that no competing interests exist.

modulation of the auditory brainstem response to one of two competing speakers by selective attention.

## Results

Assessing the brainstem's response to continuous non-repetitive speech does not allow to average over many repeated presentations of the same sound. Instead, we sought to quantify the brainstem's response to the fundamental frequency of speech. Neuronal activity in the brainstem, and in particular in the inferior colliculus, can indeed phase lock to the periodicity of voiced speech (*Skoe and Kraus, 2010*). The fundamental frequency of running speech varies over time, however, compounding a direct read-out of the evoked brainstem response.

To overcome this difficulty, we employed empirical mode decomposition (EMD) of the speech stimuli to identify an empirical mode that, at each time instance, oscillates at the fundamental frequency of the speech signal (*Huang and Pan, 2006*) (Materials and methods). This mode is a nonlinear and nonstationary oscillation with a temporally-varying amplitude and frequency that we refer to as the 'fundamental waveform' of the speech stimulus (*Figure 1a*).

We then recorded the brainstem response to running non-repetitive speech stimuli of several minutes in duration from human volunteers through scalp electrodes. We cross-correlated the obtained recording with the fundamental waveform of the speech signal (*Figure 1b*). Because the brainstem response may occur at a phase that is different from that of the fundamental waveform, we also correlated the neural signal to the Hilbert transform of the fundamental waveform that has a phase shift of 90°. The two correlations can be viewed as the real and imaginary part of a complex correlation function that can trace the brainstem response at any phase shift. The amplitude of the complex correlation informs then on the strength of the brainstem response.

Our statistical analysis showed that the peak amplitude of the complex correlation was significantly different from the noise in fourteen out of sixteen subjects (p<0.05, Materials and methods). The peak occurred at a mean latency of 9.3 ± 0.7 ms, verifying that the measured neural response resulted from the brainstem and not from the cerebral cortex (*Hashimoto et al., 1981*; *Picton et al., 1981*). Moreover, the latency agreed with that found previously regarding the brainstem's response to short repeated speech stimuli (*Skoe and Kraus, 2010*). The average value of the correlation at the peak was 0.015 ± 0.003. We checked that the response did not contain a stimulus artifact or a contribution from the cochlear microphonic, and that the latency of the response was not affected by the processing of the speech signal or of the neural response (Materials and methods; *Figure 1—figure supplement 1*). We also verified that the response did not contain further peaks at higher latencies, showing that the neural signal did not contain a measurable contribution from the cortex.

We further considered a highly simplistic model of the auditory brainstem response in which a burst of neural spikes occurred at each cycle of the fundamental waveform, at a fixed phase $\varphi$, and was then shifted in time by a certain delay $\tau$ (*Figure 2a,b*; Materials and methods). Adding noise that is realistic of neural recordings from scalp electrodes, and computing the complex correlation of the obtained signal with the fundamental waveform, showed a peak in the complex correlation at the time delay $\tau$ (*Figure 2c*). The phase $\varphi$ of the fundamental waveform at which the neural bursts occurred was obtained from the phase of the complex correlation at the time delay $\tau$. This demonstrated that the brainstem's response to continuous speech could be reliably extracted through the developed method. The response can be characterized through the latency and amplitude of the correlation's peak.

Armed with the ability to quantify the brainstem's response to running non-repetitive speech, we sought to investigate if this neural activity is affected by selective attention. Employing a well-established paradigm of attention to one of two speakers (*Ding and Simon, 2012*), we presented volunteers diotically with two concurrent speech streams of equal intensity, one by a male and another by a female voice. For parts of the speech presentation subjects attended the male voice and ignored the female voice, and *vice versa* for the remaining parts.

We quantified the brainstem's response to both the male and the female voice by extracting the fundamental waveforms of both speech signals and correlating the neural recording separately to both. We found that the latency of the response was unaffected by attention: the response to the unattended speaker occurred 0.8 ± 0.5 ms later than that to the attended speaker, which was not

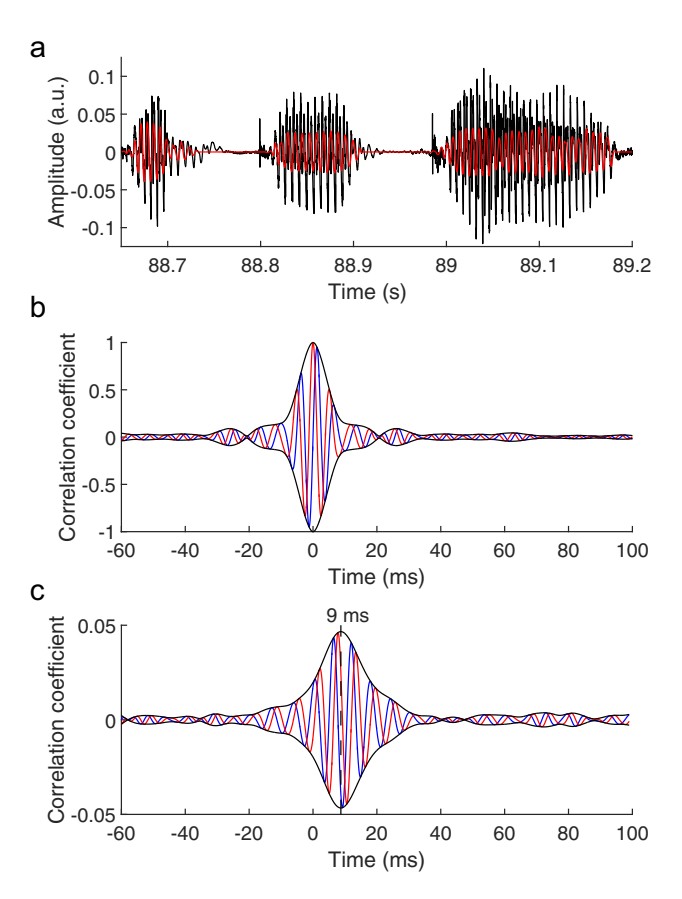

**Figure 1.** The brainstem response to running speech. (**a**) Speech (black) contains voiced parts with irregular oscillations at a time-varying fundamental frequency and higher harmonics. We extract a fundamental waveform (red) that oscillates nonlinearly and nonstationary at the fundamental frequency. (**b**) The autocorrelation of the fundamental waveform (red) peaks when the delay vanishes and oscillates at the average fundamental frequency. The cross-correlation of the fundamental waveform with its Hilbert transform (blue) can be seen as an imaginary part of the autocorrelation. The amplitude of the resulting complex cross-correlation (black) shows a life-time of a few ms. (**c**) The correlation of the speech-evoked brainstem response, recorded from one subject, to the fundamental waveform of the speech signal (red) as well as to its Hilbert transform (blue) can serve as real and imaginary parts of a complex correlation function. Its amplitude (black) peaks at a latency of 9 ms. The latency of the correlation is not altered by the processing of the speech signal or of the neural recording, and contains neither a stimulus artifact nor the cochlear microphonic (***Figure 1—figure supplement 1***).

DOI: https://doi.org/10.7554/eLife.27203.002

The following figure supplement is available for figure 1:

**Figure supplement 1.** Controls for latencies induced by signal processing as well as for the source of the measured brainstem response to running speech.

DOI: https://doi.org/10.7554/eLife.27203.003

statistically significant (p=0.2; average over the responses to the male and the female voice as well as all subjects).

In contrast, all subjects showed a larger response of the auditory brainstem, at the peak latency, to the male voice when attending rather than ignoring it (***Figure 3a***). The difference in the responses was statistically significant in nine of the fourteen subjects (p<0.05). The brainstem's response to the attended female speaker similarly exceeded that to the unattended female voice in all but one subject, with eight subjects showing a statistically-significant difference (p<0.05; ***Figure 3b***). The ratio of the brainstem's response to attended and to ignored speech, averaged over all subjects, was 1.5 ± 0.1 and 1.6 ± 0.2 for the male and for the female speaker, respectively. Both ratios were

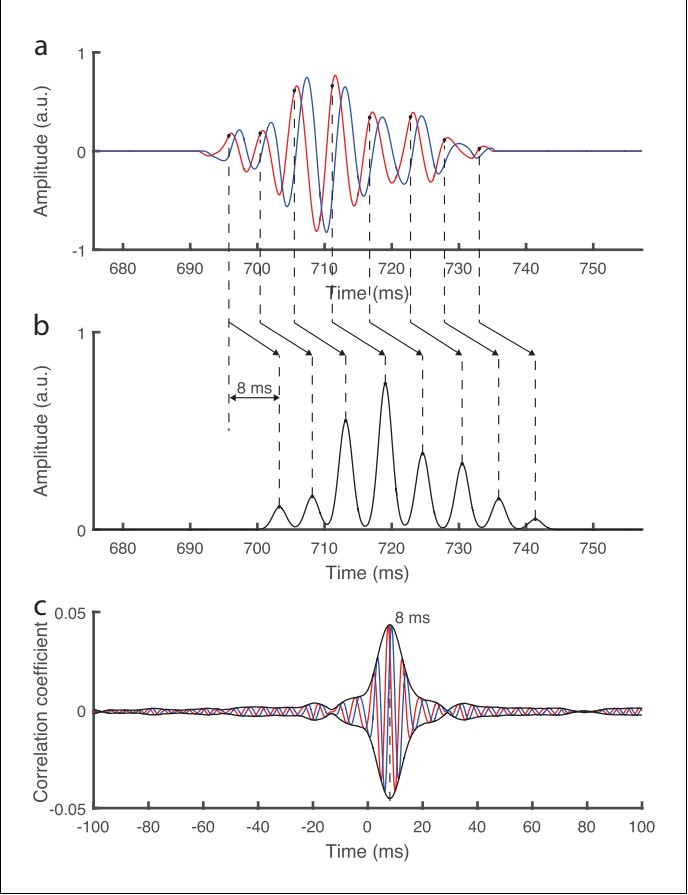

**Figure 2.** Simplistic model of the auditory brainstem response to continuous speech. (**a**) The fundamental waveform (red) as well as its Hilbert transform (blue) oscillate with a varying amplitude and frequency. (**b**) We model a simplistic brainstem response in which bursts of neural spikes occur at each cycle of the fundamental waveform, at a phase of ¼ π rad (black dots). Furthermore, all neural bursts are shifted by a temporal delay of 8 ms. (**c**) When adding realistic noise as emerges from scalp recordings, and then computing the complex correlation with the fundamental waveform as performed for the actual brainstem recording, we find a peak at the modelled delay of 8 ms. The modelled phase of ¼ π rad is obtained as the inverse phase of the complex correlation at that latency.

DOI: https://doi.org/10.7554/eLife.27203.004

significantly different from unity (p<0.001, male voice; p<0.01, female voice). The male and the female voice elicited a comparable attentional modulation: the difference between the corresponding ratios was insignificant (p=0.7). The magnitude of the brainstem's response was hence significantly enhanced through attention, and consistently so across subjects and speakers.

The auditory brainstem response to short speech stimuli has a low-pass nature: the amplitude of the response declines with increasing frequency (*Musacchia et al., 2007*; *Skoe and Kraus, 2010*).To determine if this relation held for the brainstem response to continuous speech that we measured here as well, and how it may be affected by attention, we computed the correlation between the amplitude of the brainstem response to short speech segments and the fundamental frequency of the segments (Materials and methods). We found a small but statistically significant negative correlation between the amplitude of the brainstem response and the fundamental frequency, both for the neural response to a single speaker as well as for the response to the attended and ignored speech signal of two competing speakers, evidencing the low-pass nature of the brainstem response (*Figure 3—figure supplement 1*). The correlations between the amplitude and the frequency did not differ significantly between the different conditions.

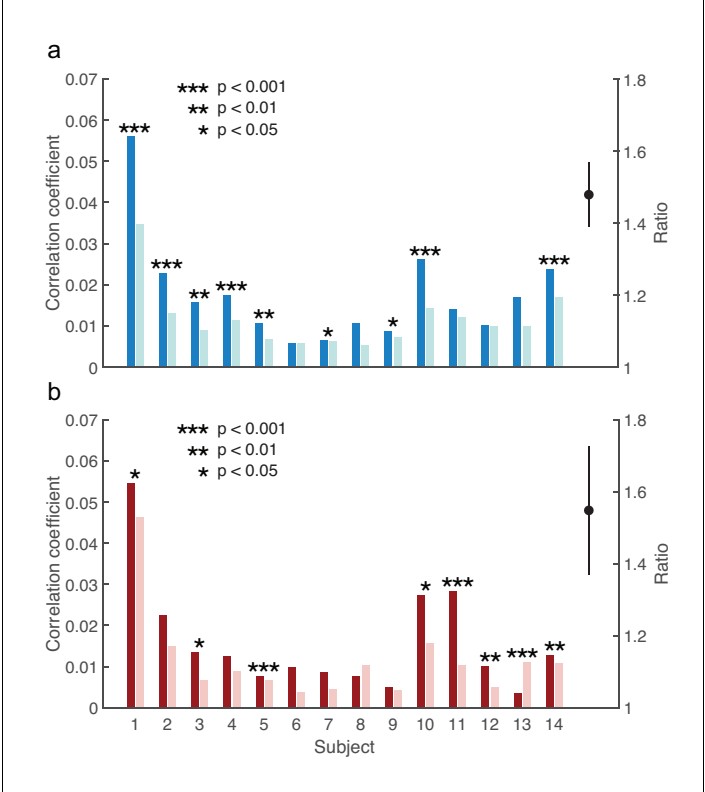

**Figure 3.** Modulation of the brainstem response to speech by selective attention. (**a**) The brainstem's response to the male speaker is larger for each subject when attending the speaker (dark blue) than when ignoring it (light blue). The average ratio of the brainstem responses to the attended and to the ignored male speaker is significantly larger than 1 (black, mean and standard error of the mean). (**b**) With the exception of subject 13, the neural response to the female voice is also larger when subjects attend to it (dark red) instead of ignoring it (light red). The average ratio of the brainstem responses to the attended and to the ignored female speaker is significantly larger than one as well (black, mean and standard error of the mean).
DOI: https://doi.org/10.7554/eLife.27203.005

The following figure supplement is available for figure 3:

**Figure supplement 1.** Correlation between the amplitude of the brainstem response and the fundamental frequency of the speech signal.
DOI: https://doi.org/10.7554/eLife.27203.006

The brainstem response to short clicks in noise can have a delay that becomes longer with increasing noise level, while the amplitude of the brainstem response declines, presumably reflecting varying contributions of auditory-nerve fibers with different spontaneous rates and from different cochlear locations (*Mehraei et al., 2016*). However, for the brainstem response to continuous speech that we measured here, we did not find a statistically-significant correlation between amplitude and latency (Materials and methods).

## Discussion

Our results show that the human auditory brainstem response to continuous speech is larger when attending than when ignoring a speech signal, and consistently so across different subjects and speakers. In particular, the strength of the phase locking of the neural activity to the pitch structure of speech is larger for an attended than for an unattended speech stream. In contrast, we did not observe a difference in the latency of this activity.

The fundamental waveform of speech that we have obtained from EMD has a temporally varying frequency and amplitude and is therefore not a simple component of Fourier analysis. While it may be obtained from short-time Fourier transform or wavelet analysis, both methods suffer from an

inherently limited time-frequency resolution that makes them inferior to the EMD analysis (*Huang and Pan, 2006*).

Because we have employed a diotic stimulus presentation in which the same acoustical stimulus was presented to each ear, the attentional modulation cannot result from a general modulation of the brainstem's activity to acoustic stimuli between the two hemispheres. Moreover, although the fundamental frequencies of the two competing speakers differ at most time points, their spectra largely overlap. The attentional modulation can therefore not result from a broad-band modulation of the neural activity either. Instead, the attentional effect must result from a modulation of the brainstem's response to the specific pitch structure of a speech stimulus.

The brainstem response to the pitch of continuous speech that we have measured can reflect a response both to the fundamental frequency of speech as well as to higher harmonics. Indeed, previous studies have found that the brainstem responds at the fundamental frequency of a speech stimulus even when that frequency itself is removed from the acoustic signal (*Galbraith and Doan, 1995*), or when it cancels out due to presentation of stimuli with opposite polarities and averaging of the obtained responses (*Aiken and Picton, 2008*). The attentional modulation of the brainstem response can thus reflect a modulation of the response to the fundamental frequency itself or to higher harmonics. Moreover, attentional modulation of higher harmonics may depend on frequency as shown recently in recordings of otoacoustic emissions from the inner ear (*Maison et al., 2001*).

The attentional modulation of the brainstem's response to the pitch of a speaker may result from an enhancement of the neural response to an attended speech signal, from the suppression of the response to an ignored speech stimulus, or from both. Further investigation into this issue may compare brainstem responses to speech when attending to the acoustical signal and when attending to a visual stimulus (*Woods et al., 1992*; *Karns and Knight, 2009*; *Saupe et al., 2009*).

The response at the fundamental frequency of speech can result from multiple sites in the brainstem (*Chandrasekaran and Kraus, 2010*). However, we observed a single peak with a width of a few ms in the correlation of the neural signal to the fundamental waveform of speech. The brainstem response to running speech that we have measured here can therefore only reflect neural sources whose latencies vary by a few ms or less from the peak latency. The neural delay of about 9 ms as well as the similarity of the speech-evoked brainstem response to the frequency-following response suggest that the main neural source may be in the inferior colliculus (*Sohmer et al., 1977*). The attentional effect that we have observed may then result from the multiple feedback loops between the inferior colliculus, the medial geniculate body and the auditory cortex (*Huffman and Henson, 1990*).

Our study provides a mathematical methodology to analyse the brainstem response to complex, real world stimuli such as speech. Since our method does not require artificial and repeated stimuli, it fosters sustained attention and avoids potential neural adaptation. This method can therefore pave the way to further explore how the brainstem contributes to the processing of complex real-world acoustic environments. It may also be relevant for better understanding and diagnosing the recently discovered cochlear neuropathy or 'hidden hearing loss' (*Kujawa and Liberman, 2009*). Because the latter alters the brainstem's activity (*Schaette and McAlpine, 2011*; *Mehraei et al., 2016*), assessing the auditory brainstem response to speech as well as its modulation by attention may further clarify the origin, prevalence and consequences of such poorly understood supra-threshold hearing loss.

## Materials and methods

### Participants

16 healthy adult volunteers aged 18 to 32, eight of which were female, participated in the study. All subjects were native English speakers and had no history of hearing or neurological impairments. All participants had pure-tone hearing thresholds better than 20 dB hearing level in both ears at octave frequencies between 250 Hz and 8 kHz. Each subject provided written informed consent. All experimental procedures were approved by the Imperial College Research Ethics Committee.

## Auditory brainstem recordings to running speech

Samples of continuous speech from a male and a female speaker were obtained from publicly available audiobooks (https://librivox.org). All samples had a duration of at least two minutes and ten seconds; some were slightly longer to end upon completion of a sentence. To construct speech samples with two competing speakers, samples from the male and from the female speaker were normalized to the same root-mean-square amplitude and then superimposed.

Participants were placed in a comfortable chair in an acoustically and electrically insulated room (IAC Acoustics, UK). A personal computer outside the room controlled audio presentation and data acquisition. Speech stimuli were presented at a sampling frequency of 44.1 kHz through a high-performance sound card (Xonar Essence STX, Asus, USA). Stimuli were delivered diotically through insert earphones (ER-3C, Etymotic, USA) at a level of 76 dB(A) SPL (A-weighted frequency response). Sound intensity was calibrated with an ear simulator (Type 4157, Brüel and Kjaer, Denmark). All subjects reported that the stimulus level was comfortable.

The response from the auditory brainstem was measured through five passive Ag/AgCl electrodes (Multitrode, BrainProducts, Germany). Two electrodes were positioned at the cranial vertex (Cz), two further electrodes were placed on the left and right mastoid processes, and the remaining electrode was positioned on the forehead to measure the ground. The impedance between each electrode and the skin was reduced to below 5 kΩ using abrasive electrolyte-gel (Abralyt HiCl, Easycap, Germany). The electrode on the left mastoid, at the cranial vertex and the ground electrode were connected to a bipolar amplifier with low-level noise and a gain of 50 (EP-PreAmp, BrainProducts, Germany). The remaining two electrodes were connected to a second identical bipolar amplifier. The output from both bipolar amplifiers was fed into an integrated amplifier (actiCHamp, BrainProducts, Germany) where it was low-pass filtered through a hardware anti-aliasing filter with a corner frequency of 4.9 kHz and sampled at 25 kHz. The audio signals were measured by the integrated amplifier as well through an acoustic adapter (Acoustical Stimulator Adapter and StimTrak, BrainProducts, Germany). The electrophysiological data were acquired through PyCorder (BrainProducts, Germany). The simultaneous measurement of the audio signal and the brainstem response from the integrated amplifier was employed to temporally align both signals to a precision of less than 40 μs, the inverse of the sampling rate (25 kHz).

## Experimental design

In the first part of the experiment, each volunteer listened to four speech samples of the female speaker only. Comprehension questions were asked at the end of each part in order to verify the subject's attention to the story.

The second part of the experiment employed eight samples of speech that contained both a male and a female voice. During the presentation of the first four samples, subjects were asked to attend either the male or the female speaker. Volunteers were then presented with the next four speech samples and asked to attend to the speaker that they had ignored earlier. Whether the subject was asked to attend first to the male or to the female voice was determined randomly for every subject. Comprehension questions were asked after each sample.

## Computation of the fundamental waveform of speech

The fundamental waveform of each speech sample with a single speaker was computed through a custom-written Matlab program (code available on Github; *Forte, 2017*; a copy is archived at https://github.com/elifesciences-publications/fundamental_waveforms_extraction). The fundamental waveform of a speech sample with two speakers followed from the two corresponding samples with a single speaker only.

First, each speech signal was downsampled to 8820 Hz, low-pass filtered at 1,500 Hz (linear-phase FIR filter, transition band 1500–1650 Hz, stopband attenuation −80 dB, passband ripple 1 dB, order 296) and time-shifted to compensate for the filter delay. Silent parts between words were identified by computing the envelope of the speech signal. Each part where the envelope was less than 10% of the maximal value found in the speech was considered silent, and the speech signal there was set to zero.

Second, the instantaneous fundamental frequency of the voiced parts of the speech signal was detected through the autocorrelation method, employing rectangular windows of 50 ms duration

with a successive overlap of 49 ms. Speech segments that yielded a fundamental frequency outside the range of 60 Hz to 400 Hz, or in which the fundamental frequency varied by more than 10 Hz between two successive windows were considered voiceless. The speech segments that corresponded to voiced speech, as well as their fundamental frequency, were thus obtained. The fundamental frequency of each segment was interpolated through a cubic spline, and varied between 100 and 300 Hz in each segment. Note that this method yields the fundamental frequency but not by itself the fundamental wavemode.

Third, the voiced speech segments were analysed through the Hilbert-Huang transform. The latter is an adaptive signal processing based on empirical basis functions and can thus be better suited for analysing nonlinear and nonstationary signals such as speech than Fourier analysis (*Huang and Pan, 2006*). The transform consists of two parts. First, empirical mode decomposition extracts intrinsic mode functions (IMFs) that satisfy two properties: (i) the numbers of extrema and zero crossings are either equal or differ by one; (ii) the mean of the upper and lower envelope vanishes. The signal follows as the linear superposition of the IMFs. Second, the Hilbert spectrum of each IMF is determined, which yields, in particular, the mode's instantaneous frequency. This analysis was performed for each short segment of voiced speech, that is, for each part of voiced speech that was preceded and followed by a pause or voiceless speech.

Fourth, the fundamental frequency of each short speech segment was compared to the instantaneous frequencies of the segment's IMFs at each individual time point. All IMFs with an instantaneous frequency that differed by less than 20% from the segment's fundamental frequency were determined, and the IMF with the largest amplitude was therefrom selected as the fundamental wavemode of that segment and at that time point (*Huang and Pan, 2006*). If no IMF had an instantaneous frequency within 20% of the fundamental frequency, or if a speech segment was unvoiced, that time point was assigned a fundamental waveform of zero. The fundamental waveforms obtained at the different time points were combined through cosine crossfading functions with a window width of 10 ms to obtain the fundamental waveform of the speech signal. The Hilbert transform of that fundamental waveform was computed as well.

To control for latency changes in the acoustic signal induced by the subsequent processing steps, and in particular by the involved frequency filtering, the cross-correlation between the original speech signal and the fundamental waveform as well as with its Hilbert transform was computed (*Figure 1—figure supplement 1a*). The cross-correlations show that the fundamental waveform has no latency change and no phase difference with respect to the original speech stimulus.

## Analysis of the auditory-brainstem response

The brainstem responses from the two measurement channels were averaged. A frequency-domain regression technique (CleanLine, EEGLAB) was used to attenuate noise from the power line in the brainstem recording. Moreover, because a voltage amplitude above 20 mV cannot result from the brainstem but represents artefacts such as spurious muscle activity, the signal was set to zero during episodes of such high voltage. The electrophysiological recording was then filtered between 100–300 Hz since the fundamental frequency of the speech was in that range (high-pass filter: linear-phase FIR filter, transition band from 90 to 100 Hz, stopband attenuation −80 dB, passband ripple 1 dB, order 6862; low-pass filter: linear-phase FIR filter, transition band 300–360 Hz, stopband attenuation −80 dB, passband ripple 1 dB, order 1054). In particular, the high-pass filter eliminated neural signals from the cerebral cortex that occur predominantly below 100 Hz. To avoid transient activity at the beginning of each speech sample, the first ten seconds of each brainstem recording in response to a speech sample were discarded. The following two minutes of data were divided into 40 epochs of a duration of 3 s each, and the remaining data were discarded, if any.

The processing of the neural signal did not induce a latency. This was confirmed by computing the cross-correlation between the processed neural response and the original signal, demonstrating a maximum correlation at zero temporal delay (*Figure 1—figure supplement 1b*).

As set out above, the first part of the experiment measured the brainstem response to running speech without background noise. For each subject and each epoch, the cross-correlation of the brainstem response with the corresponding segment of the fundamental waveform as well as with its Hilbert transform were computed. A delay of 1 ms of the acoustic signal produced by the earphones was taken into account. The two cross-correlation functions were interpreted as the real and the imaginary part of a complex correlation function. For each individual subject, the average of the

complex cross-correlation over all epochs was then computed, and the latency at which the amplitude peaked was determined.

The obtained latencies of about 9 ms affirmed that the signal resulted from the auditory brainstem and not from the cerebral cortex (*Hashimoto et al., 1981*). The latency also evidenced that the signal resulted neither from stimulus artifacts nor from the cochlear microphonic, which would occur at or near zero delay (*Skoe and Kraus, 2010*). As an additional control, the brainstem response was recorded when the earphones were near the ear, but not inserted into the ear canal, so the subject could not hear the speech signals. The recording did then not yield a measurable brainstem response (*Figure 1—figure supplement 1c*). Two presentations of the same speech stimulus, but with opposite polarities, were employed as well, and the neural response to both presentations was averaged before computing the correlation to the fundamental waveform. The correlation was identical to that obtained by a single stimulus presentation, demonstrating the absence of a stimulus artifact and of the cochlear microphonic (*Figure 1—figure supplement 1d*).

To further verify our analysis we considered a highly simplistic model of the brainstem response. In particular, we constructed a simplified hypothetical brainstem response to speech of a duration of 10 min in which neural spikes occur in bursts (*Figure 2a,b*). We described each burst by a Gaussian distribution with a width of 1 ms. Each cycle of the fundamental waveform triggered a burst, which was centered at a fixed phase $\varphi$ of the waveform and shifted by a certain time delay $\tau$. We then added an actual neural recording from a trial without sound presentation, representing realistic noise, to the simulated brainstem response, at a signal-to-noise ratio of −20 dB. Processing this simulated brainstem response through the methods described above yielded a complex correlation that peaked at the imposed time delay $\tau$ (*Figure 2c*). The phase of the complex correlation was the inverse of the phase $\varphi$ of the fundamental waveform at which the modelled brainstem response occurred; the inverse of the phase appeared due to the complex conjugate in the definition of the cross-correlation. This confirmed the validity of our methodology as well as that our processing of the neural data did not alter the temporal delay.

To determine whether the peak in the cross-correlation obtained from a given subject was significant, the values of the complex cross-correlation from the individual epochs, and at the peak latency, were analysed. Because each correlation value is an average of many measurements, it follows from the Central Limit Theorem that the complex correlations from the different epochs exhibit a two-dimensional normal distribution with a mean of zero if the measurements are randomly distributed. A one-sample Hotelling's T-squared test was therefore used to assess the significance of the complex correlation at the peak latency. Two subjects who did not show a significant correlation ($p > 0.05$) were not included in the further analysis.

The population mean and standard error of the mean of the latency were computed from the latencies of the individual subjects.

The brainstem responses to competing speakers were then analysed for each individual subject. For each epoch, the complex cross-correlation between the brainstem response and the fundamental waveform was computed, both for the fundamental waveform of the attended and for that of the unattended speaker. The corresponding complex correlation functions were averaged across epochs, and the amplitudes as well as latencies of the peaks were determined.

Statistical significance of the difference in latency of the brainstem responses to the attended and the unattended speaker, obtained from the eight samples, was tested by computing population mean as well as standard error of the mean for the differences in latencies obtained from individual subjects. A two-tailed Student's t-test was employed to test if the difference was significantly different from zero.

To control for differences in the voice of the male and the female speaker, differences in amplitude of the brainstem response to the attended and ignored male speaker were determined separately from differences in the amplitude of the brainstem response to the attended and ignored female speaker. The amplitudes of the complex cross-correlations, at the peak latencies, were computed for all epochs. A two-sample Student's t-test was then employed to test for a significant difference between the amplitude in response to the attended and the ignored speaker.

The amplitude of the brainstem response to speech can vary widely between subjects (3), due to variations such as in anatomy and scalp conductivity. The ratios of the amplitudes of the brainstem responses to attended and ignored speech, rather than the differences, were thus computed for each individual. The population mean and standard error of the mean were therefrom obtained. A

one-tailed Student's t-test assessed whether the population average of the ratio was significantly larger than unity. A two-tailed two-sample Student's t-test was employed to assess whether the ratios obtained from the responses to the male and to the female speaker were significantly different.

We assessed the correlation between the amplitude of the brainstem response and the frequency of the fundamental waveform by dividing each speech signal into 160 segments, and by dividing the corresponding neural recording analogously. We then computed the complex correlation of each segment of the neural data with the corresponding segment of the fundamental waveform, and obtained the amplitude at the peak latency. We further computed the average fundamental waveform of each segment. The correlation between amplitude and frequency was tested for statistical significance through a one-tailed Student's t-test. It was found to be significant for the brainstem response to a single speaker as well as for the response to the attended and the ignored speaker in the two-speaker stimuli ($p < 0.05$). However, none of the differences between the obtained correlation coefficients were statistically significant (two-tailed Student's t-test, $p < 0.05$).

We also computed the correlation between the amplitude and the latency of the brainstem response from short speech segments of 120 s in duration; such long segments were required to obtain a reliable assessment of the latency. We performed this analysis for the auditory brainstem response to a single speaker as well as for those to the attended and ignored speaker when subjects were presented with two competing speakers. However, none of these correlations were statistically significant (two-tailed Student's t-test, $p > 0.05$).

## Acknowledgements

We thank Steve Bell, Karolina Kluk-de Kort, Patrick Naylor, David Simpson and Malcolm Slaney for discussion as well as for comments on the manuscript. This research was supported by EPSRC grant EP/M026728/1 to TR as well as in part by the National Science Foundation under Grant No. NSF PHY-1125915.

## Additional information

### Funding

| Funder | Grant reference number | Author |
| --- | --- | --- |
| Engineering and Physical Sciences Research Council | EP/M026728/1 | Tobias Reichenbach |
| National Science Foundation | PHY-1125915 | Tobias Reichenbach |

The funders had no role in study design, data collection and interpretation, or the decision to submit the work for publication.

### Author contributions

Antonio Elia Forte, Software, Formal analysis, Validation, Investigation, Visualization, Writing—original draft, Writing—review and editing; Octave Etard, Formal analysis, Validation, Methodology, Writing—original draft, Writing—review and editing; Tobias Reichenbach, Conceptualization, Resources, Data curation, Supervision, Funding acquisition, Investigation, Methodology, Writing—original draft, Project administration, Writing—review and editing

### Author ORCIDs

Tobias Reichenbach  http://orcid.org/0000-0003-3367-3511

### Ethics

Human subjects: All subjects provided written informed consent. All experimental procedures were approved by the Imperial College Research Ethics Committee.

**Decision letter and Author response**
Decision letter https://doi.org/10.7554/eLife.27203.008
Author response https://doi.org/10.7554/eLife.27203.009

## Additional files
**Supplementary files**
• Transparent reporting form
DOI: https://doi.org/10.7554/eLife.27203.007

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
