## [Decision Letter]

Thank you for submitting your article "The human auditory brainstem response to running speech reveals a subcortical mechanism for selective attention" for consideration by *eLife*. Your article has been favorably evaluated by Andrew King (Senior Editor) and three reviewers, one of whom, Barbara G Shinn-Cunningham (Reviewer #1), is a member of our Board of Reviewing Editors. The following individual involved in review of your submission has agreed to reveal their identity: Steve Aiken (Reviewer #2).

The reviewers have discussed the reviews with one another and the Reviewing Editor has drafted this decision to help you prepare a revised submission.

Summary:

The results of this study are very intriguing, showing differences in auditory steady-state responses (ASSRs) that are specific to which of two speech streams a listener is attending. Although the correlations between the stimulus and response are small, the peak of the cross-correlation function was significant in most of the subjects, as was the effect of attention. The writing is generally clear and concise.

All three reviewers thought that the conclusion that brainstem-generated responses are modulated by attentional focus, if properly justified, is novel and noteworthy. While many aspects of this paper are intriguing, there are questions that affect the interpretation. These technical issues and concerns must be addressed adequately for the study to be acceptable for publication in *eLife*.

Essential revisions:

1) For any response like this, including the more-common FFR (with a steady-state constant-frequency acoustic signal), observations are a mixture of IC responses and other responses (perhaps in thalamus – but also lower responses such as the cochlear microphonic). For the same relative delays and magnitudes of the responses, these different responses will add in different phases, depending on their frequency. Unlike with the FFR, here, the frequency is changing from moment to moment. This will lead to different cancellation / summation at different frequencies that likely result in different peak delays. Only if there is a single truly dominant source in the mixture will the peak delay be at a fixed delay independent of frequency.

Attention effects in FFRs have been suggested to be due to the involvement of the cortex (the work of Emma Holmes presented at ARO 2017). While the delay of 10 ms relative to the stimulus calls into question a dominant role for the cortex in the present study (see above), it is also possible that the attention effect is mediated through olivocochlear inhibition.

The stats show that attention is changing the observed responses. The question is just where these responses are coming from. Given that the observed response is a mixture, further analysis is warranted to tease apart whether the effects are due to a single dominant source with a fixed delay that is modulated by attention, or whether higher-level sources, which are modulated by attention, cause different summation / cancellation effects depending on attentional focus.

2) The latency reported, 10.3ms, is greater than is usually attributed to a brainstem response. The latency of the largest peak of the click-evoked ABR (usually attributed to inferior colliculus) is usually assumed to be ~5ms or somewhat greater for lower-frequency stimuli (Don and Eggermont, 1978).

Often, the delay in an ASSR is longer (presumably in part because it is a mixture of neural sources as noted above). The authors say that their estimate agrees with that reported by Skoe and Kraus (2010) for speech, but the value reported in that paper is actually 7-8ms. Figure 1 of that paper, where that value is mentioned, illustrates it as the amount by which the response *precedes* the stimulus (!), so that source itself may have some methodological problems.

Because the latency is the primary fact used to conclude that the induced attentional changes are from brainstem, this issue is very important to consider and discuss.

3) The latency is calculated as a cross-correlation between electrode signals and a "fundamental waveform" defined as a "nonlinear oscillation" derived from the speech by empirical mode decomposition (EMD). EMD is attractive but not very well defined or theoretically grounded; as far as we can tell, it just extracts an approximation of the fundamental Fourier component. The Hilbert transform is calculated, and both the waveform and its Hilbert transform are cross-correlated with the EEG to obtain a "complex correlation function", the amplitude of which peaks at a latency of 10.3ms. The rationale for introducing this Hilbert component is not clear, as it would seem more straightforward to correlate simply with the speech waveform (or its "fundamental waveform). The "amplitude of the complex CC" has a wider peak than the raw CC.

Please explain and justify the analysis more clearly.

4) An accurate estimate of latency is crucial for saying that the response reflects the brainstem. Temporal alignment between audio and EEG may be affected by acoustic delay in the earphones (not specified, possibly ~1ms for ER 3), as well as the signal processing of the inputs and of the brain measures.

Audio is down-sampled (interpolation filter unspecified), filtered by a FIR of order 296 (IR temporal extent 33.4 ms), time-shifted to "compensate for delay" of the FIR, processed by the EMD algorithm, and finally by the Hilbert transform. The Hilbert transform is presumably performed by applying an STFT to a window of unspecified duration. It involves a 90-degree phase shift that translates (for the quasi-sinusoidal fundamental wave) to a frequency-dependent time shift of up to 2.5ms at 100Hz.

On the EEG side, the signal is processed by a frequency-domain method (ClearLine) to attenuate 50Hz and (presumably) harmonics. The possibility that this might affect the fundamental waveform (its time-varying frequency falls in this range) is not discussed. The EEG is filtered by a cascade of FIR filters of order 6862 and 1054 (IR lengths 274ms and 42ms) before correlation with the audio-based signal. There are clearly many stages at which a latency mismatch could arise, and the fact that this is not acknowledged or addressed (for example by calibration) is troubling.

The peak value of the cross-correlation function shown in Figure 1, 0.05, seems rather high given that the ABR is supposed to have very low SNR. Similar values (0.05 – 0.1) have been reported for cross-correlation with filtered cortical responses that are supposed to have a much better SNR. The shape of the function is also somewhat intriguing because it is approximately symmetrical and extends to negative lag (i.e., there is a noncausal relationship between the input and the neural response). This suggests that it is largely determined by temporal smearing in the processing, for example due to convolution with the various unspecified filter kernels.

Please carefully outline how the acoustical and neural signal processing affects the estimated latency of the neural responses.

5) From the description of methods, it would seem that the stimulus is always presented with the same polarity. Electrode signals measured in that case are likely to be dominated by cochlear microphonic or possibly even cross-talk from the earphone drivers or cables. There is no reason to believe that they are from brainstem, except possibly latency (and as discussed above, it is unclear how good an indicator that is of brainstem activity).

Please include some calibration or analysis to verify that electrical artifact is not a significant factor in your findings.

6) While the speech presentation levels were relatively high, individual high-frequency harmonics would be relatively low in level given the low-pass characteristic of speech signals. Attentional modulation of cochlear gain has been shown to be frequency-specific (e.g., Maison et al., Psychophysiol 2001) and certainly could be specific to a harmonic complex.

It would be useful to include this point in the Discussion.

[Editors' note: further revisions were requested prior to acceptance, as described below.]

Thank you for submitting your article "The human auditory brainstem response to running speech reveals a subcortical mechanism for selective attention" for consideration by *eLife*. Your article has been favorably evaluated by Andrew King (Senior Editor) and three reviewers, one of whom, Barbara G Shinn-Cunningham (Reviewer #1), is a member of our Board of Reviewing Editors. The following individual involved in review of your submission has agreed to reveal their identity: Steve Aiken (Reviewer #2).

The reviewers have discussed the reviews with one another and the Reviewing Editor has drafted this decision to help you prepare a revised submission. It is not normally *eLife* policy to allow a paper to go through multiple rounds of reviews, so this will be the last opportunity to revise the manuscript before a final decision is made.

Summary:

This paper provides evidence for attentional modulation in neural responses measured in response to running speech. The approach is relatively novel and the findings interesting.

The revision has gone some way to addressing the concerns raised by the reviewers. However, several questions still need to be answered. The reviewers also provide suggestions for how to strengthen the argument. The controls added here would definitely strengthen the paper.

Essential revisions:

1) The paper continues to over-emphasize that the measured responses are from the brainstem. The evidence shows a clear attentional effect; however, the claim that the observed effects are absolutely from the brainstem is still problematic. If cortical contributions cannot be ruled out, it would be presumptuous to conclude that this is an attentional effect in the brainstem.

The authors should make their case more persuasively. One approach to enhance this argument is to conduct further analysis of their present data. If cortical contributions are indeed responsible for the attentional effects, one would expect to find a positive correlation between peak latency and amplitude in the attended conditions. One might also expect such effects to be stronger for segments with lower fundamental frequencies (e.g., < 200 Hz) given cortical phase-locking limits.

The authors should test for these possibilities. Of interest would be any relationship between peak cross-correlation latency and peak cross-correlation amplitude for attended streams, and between peak cross-correlation amplitude and segment fundamental frequency (also for attended vs. unattended streams). Such an analysis might lead to a more nuanced understanding of the data or conversely add weight to the conclusions.

2) The cross-correlation functions between speech and "fundamental wave" and between raw and processed EEG) address earlier concerns about the effect of processing on latency estimates. Still, an end-to-end calibration would be even more convincing. This does not require recording new data, but rather running simulated data through the processing pipeline: (a) formulate a simple speech-to-neural response model based on the conclusions the authors believe follow from their results, (b) add background EEG signal (e.g., from recorded data shifted in time), (c) run the analysis, (d) check whether latencies conform to what is expected based on the model. Given the importance of the timing of the responses to the argument that the effects are subcortical, the extra effort is worthwhile: the EEG processing pipeline involves many convolutional stages that the authors still do not fully characterize (are all filters zero-phase?).

---

## [Author Response]

Essential revisions:1) For any response like this, including the more-common FFR (with a steady-state constant-frequency acoustic signal), observations are a mixture of IC responses and other responses (perhaps in thalamus – but also lower responses such as the cochlear microphonic). For the same relative delays and magnitudes of the responses, these different responses will add in different phases, depending on their frequency. Unlike with the FFR, here, the frequency is changing from moment to moment. This will lead to different cancellation / summation at different frequencies that likely result in different peak delays. Only if there is a single truly dominant source in the mixture will the peak delay be at a fixed delay independent of frequency.Attention effects in FFRs have been suggested to be due to the involvement of the cortex (the work of Emma Holmes presented at ARO 2017). While the delay of 10 ms relative to the stimulus calls into question a dominant role for the cortex in the present study (see above), it is also possible that the attention effect is mediated through olivocochlear inhibition.The stats show that attention is changing the observed responses. The question is just where these responses are coming from. Given that the observed response is a mixture, further analysis is warranted to tease apart whether the effects are due to a single dominant source with a fixed delay that is modulated by attention, or whether higher-level sources, which are modulated by attention, cause different summation / cancellation effects depending on attentional focus.

We have now added additional controls for a potential stimulus artifact as well as for a contribution from the cochlear microphonic. In particular, we have measured the brainstem response to a speech signal as well as to the same signal with reversed polarity. The averaged neural response shows the same peak in its complex correlation to speech as the response to a single speech presentation. This evidences the absence of a stimulus artifact as well as of a measurable cochlear microphonics.

The brainstem response at the fundamental frequency of speech can indeed result from multiple sites in the brainstem. However, because we observe a single peak with a width of a few ms in the correlation of the neural response to the fundamental waveform of speech, the brainstem response to speech that we describe here cannot reflect sources whose latencies vary by more than a few ms from the mean latency. The latency of about 9 ms confirms the absence of a cortical contribution and hints at a neural origin in the inferior colliculus. The inferior colliculus is indeed connected to the auditory cortex through multiple segregated feedback loops, involving the olivocochlear efferent system, that likely mediate the attentional modulation that we report here.

These important issues are now discussed at different places in the revised manuscript, namely in the Results section, in a new paragraph in the Discussion section, and in three new paragraphs in the Materials and methods section. The additional data showing the control for a stimulus artifact and the cochlear microphonic is laid out in a new supplement to Figure 1.

2) The latency reported, 10.3ms, is greater than is usually attributed to a brainstem response. The latency of the largest peak of the click-evoked ABR (usually attributed to inferior colliculus) is usually assumed to be ~5ms or somewhat greater for lower-frequency stimuli (Don and Eggermont, 1978).Often, the delay in an ASSR is longer (presumably in part because it is a mixture of neural sources as noted above). The authors say that their estimate agrees with that reported by Skoe and Kraus (2010) for speech, but the value reported in that paper is actually 7-8ms. Figure 1 of that paper, where that value is mentioned, illustrates it as the amount by which the response precedes the stimulus (!), so that source itself may have some methodological problems.Because the latency is the primary fact used to conclude that the induced attentional changes are from brainstem, this issue is very important to consider and discuss.

Figure 1 of the publication by Skoe and Kraus (2010) reports indeed a delay of 7-8 ms. However, this is only one example of a measured response, and the authors state in the same publication that the measured latencies occur at a delay of 6 – 10 ms (page 15). As pointed out by the reviewers below, the earphones that we have employed introduce a delay of 1 ms that we had not accounted for earlier. We have now corrected our measurements for this delay, and the latencies that we obtain are now on average 9.3 ms, which is consistent with the results summarized in the review by Skoe and Kraus (2010). We have modified the manuscript and Discussion to reflect this important point.

Regarding the latency reported in Figure 1 of the publication by Skoe and Kraus (2010), the acoustic stimulus has been shifted to a later time by 7 – 8 ms with respect to the neural response to maximize the visual coherence between both signals. The neural response reported in this Figure therefore occurs 7 – 8 ms *after* the acoustic stimulus.

3) The latency is calculated as a cross-correlation between electrode signals and a "fundamental waveform" defined as a "nonlinear oscillation" derived from the speech by empirical mode decomposition (EMD). EMD is attractive but not very well defined or theoretically grounded; as far as we can tell, it just extracts an approximation of the fundamental Fourier component. The Hilbert transform is calculated, and both the waveform and its Hilbert transform are cross-correlated with the EEG to obtain a "complex correlation function", the amplitude of which peaks at a latency of 10.3ms. The rationale for introducing this Hilbert component is not clear, as it would seem more straightforward to correlate simply with the speech waveform (or its "fundamental waveform). The "amplitude of the complex CC" has a wider peak than the raw CC.Please explain and justify the analysis more clearly.

The fundamental frequency of running speech varies with time. Moreover, the amplitude of the voiced parts of speech also varies greatly, between zero for voiceless speech and a maximal value for voiced speech. While these variations may be ignored when investigating responses to short repeating stimuli, where the frequency fluctuations are limited due to the short duration and the amplitude may be approximately constant, the variations cannot be ignored in running speech where the fundamental frequency may vary over an octave and the amplitude varies widely. Fourier analysis decomposes a signal into sinusoidal oscillations at different, but constant, frequency and amplitude. A single component extracted from Fourier analysis can thus not represent the fundamental waveform. Furthermore, classical time-frequency analysis methods such as the Short Time Fourier Transform and wavelet-based methods are not well suited for this task either due to their inherently limited time-frequency resolution (Huang and Pan 2006). EMD, in contrast, decomposes a signal into empirical modes that are nonlinear and non-stationary oscillations with time-varying frequency and amplitude with no inherent limitation to time-frequency resolution. A previous study by Huang and Pan (2006) has shown that one of these modes oscillates at the fundamental frequency. We therefore identify this mode with the fundamental waveform and employ EMD to extract it. As the reviewers state, a precise theoretical grounding of EMD is still lacking, but it is a well-defined constructive procedure that is increasingly used for analyzing nonlinear and non-stationary oscillations. We now comment further on this issue in the Discussion section of our revised manuscript.

As the reviewers remark, the correlation of the neural response to the fundamental waveform has a narrower peak than the amplitude of the complex correlation. However, this narrower correlation is partly due to phase mismatch, and not to a lower correlation per se. Indeed, the negative correlation at a few ms before and after the peak shows that the brainstem response is then in antiphase, but still correlated to, the fundamental waveform. More generally, the brainstem response can occur at an arbitrary phase with respect to the fundamental waveform. We therefore compute the correlation of the neural signal to the Hilbert transform of the fundamental waveform as well. The Hilbert transform is a version of the fundamental waveform that is phase shifted by 90˚. By interpreting both correlations as the real and imaginary part of a complex correlation, we can track a brainstem response at an arbitrary phase delay. This analysis is reminiscent of Fourier analysis at a particular frequency where the correlation of a signal to a cosine and a sine function are interpreted as real and imaginary part of a complex Fourier coefficient, the phase of which represents the phase delay of the signal with respect to the cosine function. We now explain this important point in the Results section.

4) An accurate estimate of latency is crucial for saying that the response reflects the brainstem. Temporal alignment between audio and EEG may be affected by acoustic delay in the earphones (not specified, possibly ~1ms for ER 3), as well as the signal processing of the inputs and of the brain measures.Audio is down-sampled (interpolation filter unspecified), filtered by a FIR of order 296 (IR temporal extent 33.4 ms), time-shifted to "compensate for delay" of the FIR, processed by the EMD algorithm, and finally by the Hilbert transform. The Hilbert transform is presumably performed by applying an STFT to a window of unspecified duration. It involves a 90-degree phase shift that translates (for the quasi-sinusoidal fundamental wave) to a frequency-dependent time shift of up to 2.5ms at 100Hz.On the EEG side, the signal is processed by a frequency-domain method (ClearLine) to attenuate 50Hz and (presumably) harmonics. The possibility that this might affect the fundamental waveform (its time-varying frequency falls in this range) is not discussed. The EEG is filtered by a cascade of FIR filters of order 6862 and 1054 (IR lengths 274ms and 42ms) before correlation with the audio-based signal. There are clearly many stages at which a latency mismatch could arise, and the fact that this is not acknowledged or addressed (for example by calibration) is troubling.The peak value of the cross-correlation function shown in Figure 1, 0.05, seems rather high given that the ABR is supposed to have very low SNR. Similar values (0.05 – 0.1) have been reported for cross-correlation with filtered cortical responses that are supposed to have a much better SNR. The shape of the function is also somewhat intriguing because it is approximately symmetrical and extends to negative lag (i.e., there is a noncausal relationship between the input and the neural response). This suggests that it is largely determined by temporal smearing in the processing, for example due to convolution with the various unspecified filter kernels.Please carefully outline how the acoustical and neural signal processing affects the estimated latency of the neural responses.

We are grateful to the reviewers for pointing out the acoustic delay in the earphones. Manufacture specifications state that the earphones indeed introduce a delay of 1 ms. As explained above, we have now adjusted for this delay, and the average delay of the brainstem response to speech that we have measured is now 9.3 ms. The figures and the text of our revised manuscript have been modified accordingly.

We have taken care that the various steps involved in the processing of the acoustic and the neural signals do not affect the latencies. In particular, the down-sampling of the audio signal was carried out using the Matlab resample function, which resamples an input sequence by applying an antialiasing FIR lowpass filter and compensating for the delay introduced by the filter. The Hilbert transform did not involve a STFT, and the other frequency filters were time-compensated to not produce a latency shift either.

Due to the importance of potential latency shifts introduced by signal processing, we have investigated this issue further through computing cross-correlations of the processed and the unprocessed signals. The obtained data are presented in the novel supplement to Figure 1, and are discussed in the Results section as well as in two new paragraphs in the Materials and methods section. Our analysis shows that there is no latency shift between the computed fundamental waveform and the original speech signal, and neither is there a latency shift between the processed and the original neural recording.

The brainstem response is indeed much weaker than cortical responses. We therefore require long recordings of 10 minutes in duration to achieve a good SNR. Nonetheless, the average of the correlation across the different subjects is only 0.015 ± 0.003 as we now report in the manuscript. Moreover, we employed passive electrodes in combination with a specialized bipolar preamplifier with a very high common-mode rejection; this helps to achieve a high SNR. Measurements of cortical responses often employ active electrodes that have a lower SNR than passive electrodes as well as a main amplifier with a lower common-mode rejection than our specialized bipolar preamplifier, which decreases the SNR of the so-obtained cortical responses further.

As the reviewers remark, the peak of the correlation is approximately symmetric and extends to negative delays. The shape of the peak reflects indeed the autocorrelation of the speech signal (Figure 1) which is symmetric. The peak of the complex correlation of the neural response to the stimulus is wider than that of the autocorrelation since it is not only affected by the self-similarity of the fundamental waveform at short times, but also by that of the neural response as well as by noise. The causality between the acoustic signal and the neural response is established by the positive delay of the peak of the correlation. The extension of the lower-delay part of this peak to negative times does, in contrast, not reflect a violation of causality. Indeed, the peak in the autocorrelation of the fundamental waveform extends to negative times as well but it is clearly the lag at the peak (0 ms) that determines the casual relationship.

5) From the description of methods, it would seem that the stimulus is always presented with the same polarity. Electrode signals measured in that case are likely to be dominated by cochlear microphonic or possibly even cross-talk from the earphone drivers or cables. There is no reason to believe that they are from brainstem, except possibly latency (and as discussed above, it is unclear how good an indicator that is of brainstem activity).Please include some calibration or analysis to verify that electrical artifact is not a significant factor in your findings.

We agree that special care needs to be taken to ensure that the measured response is free from stimulus artifacts.

The influence of stimulus artifacts or of the cochlear microphonics is unlikely since both would manifest at zero latency where we do not observe a measurable response. However, we have now carried out two additional tests as suggested by the reviewers. The results are reported in a new supplement to Figure 1 and are discussed in the Results section as well as in a new paragraph in the Materials and methods section.

First, we have recorded brainstem responses when the earphones were not inside the ear canal of a subject but close to the ear, so that the subject could not hear the stimulus. We do then not obtain a measurable brainstem response to speech, evidencing the absence of stimulus artifacts. Second, we have played the same story to a subject twice, the second time with inverted polarity. We have then averaged the neural response to both stimulus presentations and performed the correlation analysis on the average. This procedure removes putative stimulus artifacts as well as the cochlear microphonic. We obtain the same correlation as when we analyze the response to a single stimulus, demonstrating that the response reflects neither a stimulus artifact nor the cochlear microphonic.

6) While the speech presentation levels were relatively high, individual high-frequency harmonics would be relatively low in level given the low-pass characteristic of speech signals. Attentional modulation of cochlear gain has been shown to be frequency-specific (e.g., Maison et al., Psychophysiol 2001) and certainly could be specific to a harmonic complex.It would be useful to include this point in the Discussion.

The response at the fundamental frequency that we have measured can indeed result from higher harmonics as we had already discussed in our manuscript. As the reviewer remarks, attentional modulation may depend on the frequency of the harmonics. We have now included this valuable point in the Discussion section of our revised manuscript.

[Editors' note: further revisions were requested prior to acceptance, as described below.]

Essential revisions:1) The paper continues to over-emphasize that the measured responses are from the brainstem. The evidence shows a clear attentional effect; however, the claim that the observed effects are absolutely from the brainstem is still problematic. If cortical contributions cannot be ruled out, it would be presumptuous to conclude that this is an attentional effect in the brainstem.The authors should make their case more persuasively. One approach to enhance this argument is to conduct further analysis of their present data. If cortical contributions are indeed responsible for the attentional effects, one would expect to find a positive correlation between peak latency and amplitude in the attended conditions. One might also expect such effects to be stronger for segments with lower fundamental frequencies (e.g., < 200 Hz) given cortical phase-locking limits.The authors should test for these possibilities. Of interest would be any relationship between peak cross-correlation latency and peak cross-correlation amplitude for attended streams, and between peak cross-correlation amplitude and segment fundamental frequency (also for attended vs. unattended streams). Such an analysis might lead to a more nuanced understanding of the data or conversely add weight to the conclusions.

Cortical contributions can be ruled out due to their longer latency. Hashimoto et al. (1981), for instance, employed recordings in neurosurgical patients from different parts of the brainstem as well as the cortex during sound stimulation. They found that the cortex did not contribute to any scalp-recorded potential with a delay of 10 ms or less. Picton et al. (1981) found that any auditory-evoked scalp-recorded potential with a latency of 15 ms or less originates from the cochlea or the brainstem. The neural response to the fundamental frequency of speech that we measure here has a latency of 9.3 ms which shows that this response does not contain a cortical contribution.

Our obtained latency is now further validated through a toy model of the brainstem response that we have included in our revised manuscript as set out below. Moreover, we have now included the new panel (E) to Figure 1—figure supplement 1 that shows the correlation between the neural signal and the complex fundamental waveform up to a latency of 700 ms, evidencing that there is no peak except for the one at 9 ms, and therefore no measurable contribution from the cerebral cortex. We now also point out that we assess the brainstem response through the amplitude of the complex cross-correlation at the peak amplitude. Even if there was a small cortical contribution to the neural response it would occur at a higher latency and would thus not affect our results regarding attention.

We now make this case more persuasively in our revised manuscript and cite the two important references described above. We now write:

"The peak occurred at a mean latency of 9.3 ± 0.7 ms, verifying that the measured neural response resulted from the brainstem and not from the cerebral cortex (Hashimoto et al. 1981; Picton et al. 1981)."

We have thereby added the new references to Hashimoto et al. (1981) as well as to Picton et al. (1981).

We have also followed the valuable suggestion to investigate correlations between amplitude and latency of the brainstem response. We find, however, no statistically significant correlation. In particular, we have analysed the correlation between amplitude and latency of short segments of the recordings and of the corresponding speech signal. We have performed this analysis for several conditions, namely for the brainstem response to a single speaker as well as for the neural response to both the attended and the ignored speaker of the two-speaker stimulus. For each speech stimulus we have also determined the parts with a fundamental frequency that is lower than the mean frequency, as well as the parts with a fundamental frequency above the mean, and considered each condition separately. However, in neither condition did we find a statistically significant correlation between the amplitude and the latency.

We would like to point out that this does not necessarily mean that there is no correlation between amplitude and latency, but only that we were not able to find such a correlation from our data. Indeed, a recent study by Mehraei et al. (J. Neurosci. 2016), finds a correlation between amplitude and latency of wave V of the brainstem response to clicks in noise when the noise level is varied. This correlation presumably arises from the varying contribution of auditory-nerve fibers with different thresholds and from different parts of the cochlea. Since speech contains varying contributions other than the fundamental waveform, it is therefore conceivable that a similar correlation exists in the brainstem response to continuous speech that we describe, but that the effect is too small to emerge from our data. On the contrary, this implies that even if such an effect was found, it alone would not allow to differentiate between brainstem and cortical contributions to the observed response.

We have also followed the important suggestion to investigate a correlation between the amplitude of the brainstem response and the fundamental frequency of the speech signal. We have done this through considering short speech segments and the corresponding neural data. This analysis has been performed for the brainstem response to a single speaker as well as for that to the attended and the ignored speaker of the two-speaker stimulus. We find small negative and statistically significant correlations between amplitude and fundamental frequency in all conditions. This concurs with previous results that have shown a low-pass nature of the brainstem response (e.g. Musacchia, et al. (2007)). We find, however, no statistically significant difference between the correlation coefficients in the different conditions. In particular, the correlation obtained from the attended speaker is not significantly larger than that obtained from the ignored speaker.

We now describe these findings in detail in our revised manuscript, and are convinced that they will trigger further studies into the important issues of the precise neural mechanisms that govern the brainstem response to continuous speech.

In particular, we have added the new Figure 3—figure supplement 1 in which we show the correlation between the amplitude of the brainstem response and the fundamental frequency of the speech signal.

In the Results section, we write:

"The auditory brainstem response to short speech stimuli has a low-pass nature: the amplitude of the response declines with increasing frequency (Musacchia et al. 2007; Skoe & Kraus 2010). […] However, for the brainstem response to continuous speech that we measured here, we did not find a statistically-significant correlation between amplitude and latency (Materials and methods).”

In the Materials and methods section, we write:

"We assessed the correlation between the amplitude and the frequency of the fundamental waveform by dividing each speech signal into 160 segments (3 s duration each), and by dividing the corresponding neural recording analogously. […] However, none of these correlations were statistically significant (two-tailed Student's t-test, p > 0.05).”

2) The cross-correlation functions between speech and "fundamental wave" and between raw and processed EEG) address earlier concerns about the effect of processing on latency estimates. Still, an end-to-end calibration would be even more convincing. This does not require recording new data, but rather running simulated data through the processing pipeline: (a) formulate a simple speech-to-neural response model based on the conclusions the authors believe follow from their results, (b) add background EEG signal (e.g., from recorded data shifted in time), (c) run the analysis, (d) check whether latencies conform to what is expected based on the model. Given the importance of the timing of the responses to the argument that the effects are subcortical, the extra effort is worthwhile: the EEG processing pipeline involves many convolutional stages that the authors still do not fully characterize (are all filters zero-phase?).

We agree that a simulated brainstem response, processed through our pipeline and analysed with our methodology, will strengthen our argument in two ways. First, it will show that our methodology is able to accurately extract both phase shift *and* time delay of the brainstem signal. Second, it will provide a verification that our processing, such as through the involved filtering, does not introduce an additional time delay.

We have therefore followed this very valuable suggestion and considered a toy model for the brainstem response. In a highly simplistic fashion, the brainstem response is thereby modelled as a series of bursts of spikes. Each cycle of the fundamental waveform triggers a burst that occurs at a fixed phase of the fundamental waveform. All bursts are then shifted by a certain fixed time delay. We further added noise that is realistic for scalp recordings, and then analysed the resulting modelled signal through our processing methodology. We then process the simulated response through our signal-processing pipeline. To clarify, all the filters that we used to process the data are linear phase FIR filter, and we systematically compensated for their delay.

We find a complex correlation that performs as expected: the peak of the amplitude occurs at the time delay of the modelled response, and the phase yields the phase of the fundamental waveform at which the bursts occur. This verifies that our methodology is indeed able to compute both phase shift *and* time delay of the brainstem response. It also verifies that the processing that we employ for the neural recording does not yield an additional time delay.

We now describe this simplistic model and the important findings that follow from it in our revised manuscript. In particular, in the Results section, we now write:

"We further considered a highly simplistic model of the auditory brainstem response in which a burst of neural spikes occurred at each cycle of the fundamental waveform, at a fixed phase φ, and was then shifted in time by a certain delay τ (Figure 2; Materials and methods). […] This demonstrated that the brainstem's response to continuous speech could be reliably extracted through the developed method."

In the Materials and methods section, we elaborate further:

"To further verify our analysis we considered a highly simplistic model of the brainstem response. […] This confirmed the validity of our methodology as well as that our processing of the neural data did not alter the temporal delay.”

Our simplistic brainstem response, together with the resulting complex correlation to the fundamental waveform, is illustrated in the new Figure 2.

We now also clarify in our revised manuscript that we employed linear phase FIR filter, and that we systematically compensated for their delay (subsection “Computation of the fundamental waveform of speech”, second paragraph and subsection “Analysis of the auditory-brainstem response”, first paragraph).